# Machine-learning-based prediction of disability progression in multiple sclerosis: An observational, international, multi-center study

Edward De Brouwer[1⊙], Thijs Becker[2,3⊙], Lorin Werthen-Brabants[4], Pieter Dewulf[5], Dimitrios Iliadis[5], Cathérine Dekeyser[6,7,8], Guy Laureys[6,7], Bart Van Wijmeersch[9,10], Veronica Popescu[9,10], Tom Dhaene[4], Dirk Deschrijver[4], Willem Waegeman[5], Bernard De Baets[5], Michiel Stock[5,11], Dana Horakova[12], Francesco Patti[13], Guillermo Izquierdo[14], Sara Eichau[14], Marc Girard[15], Alexandre Prat[15], Alessandra Lugaresi[16], Pierre Grammond[17], Tomas Kalincik[18,19], Raed Alroughani[20], Francois Grand'Maison[21], Olga Skibina[22], Murat Terzi[23], Jeannette Lechner-Scott[24], Oliver Gerlach[25,26], Samia J. Khoury[27], Elisabetta Cartechini[28], Vincent Van Pesch[29], Maria José Sà[30], Bianca Weinstock-Guttman[31], Yolanda Blanco[32], Radek Ampapa[33], Daniele Spitaleri[34], Claudio Solaro[35], Davide Maimone[36], Aysun Soysal[37], Gerardo Iuliano[38], Riadh Gouider[39], Tamara Castillo-Triviño[40], José Luis Sánchez-Menoyo[41], Guy Laureys[42], Anneke van der Walt[43], Jiwon Oh[44], Eduardo Aguera-Morales[45], Ayse Altintas[46], Abdullah Al-Asmi[47], Koen de Gans[48], Yara Fragoso[49], Tunde Csepany[50], Suzanne Hodgkinson[51], Norma Deri[52], Talal Al-Harbi[53], Bruce Taylor[54], Orla Gray[55], Patrice Lalive[56], Csilla Rozsa[57], Chris McGuigan[58], Allan Kermode[59], Angel Pérez Sempere[60], Simu Mihaela[61], Magdolna Simo[62], Todd Hardy[63], Danny Decoo[64], Stella Hughes[65], Nikolaos Grigoriadis[66], Attila Sas[67], Norbert Vella[68], Yves Moreau[1], Liesbet Peeters[3,10*]

1 ESAT-STADIUS, KU Leuven, Belgium, 2 I-Biostat, Hasselt University, Belgium, 3 Data Science Institute, Hasselt University, Belgium, 4 SUMO, IDLAB, Ghent University - imec, Belgium, 5 KERMIT, Department of Data Analysis and Mathematical Modelling, Ghent University, Belgium, 6 Department of Neurology, Ghent University, Belgium, 7 4 Brain, Ghent University, Belgium, 8 Biomedical Research Institute, Hasselt University, Belgium, 9 Noorderhart ziekenhuizen Pelt, Belgium, 10 Universitair MS Centrum Hasselt-Pelt, Belgium, 11 Biobix, Department of Data Analysis and Mathematical Modelling, Ghent University, Belgium, 12 Charles University in Prague and General University Hospital, Prague, Czech Republic, 13 Department of Medical and Surgical Sciences and Advanced Technologies, GF Ingrassia, Catania, Italy, 14 Hospital Universitario Virgen Macarena, Sevilla, Spain, 15 CHUM and Université de Montreal, Montreal, Canada, 16 IRCCS Istituto delle Scienze Neurologiche di Bologna, Bologna, Italia and Dipartimento di Scienze Biomediche e Neuromotorie, Università di Bologna, Bologna, Italia, 17 CISSS Chaudière-Appalache, Levis, Canada, 18 Melbourne MS Centre, Department of Neurology, Royal Melbourne Hospital, Melbourne, Australia, 19 CORe, Department of Medicine, University of Melbourne, Melbourne, Australia, 20 Amiri Hospital, Sharq, Kuwait, 21 Neuro Rive-Sud, Quebec, Canada, 22 Box Hill Hospital, Melbourne, Australia, 23 19 Mayis University, Samsun, Turkey, 24 University Newcastle, Newcastle, Australia, 25 Academic MS Center Zuyderland, Department of Neurology, Zuyderland Medical Center, Sittard-Geleen, The Netherlands, 26 School for Mental Health and Neuroscience, Maastricht University, Maastricht, The Netherlands, 27 American University of Beirut Medical Center, Beirut, Lebanon, 28 Azienda Sanitaria Unica Regionale Marche - AV3, Macerata, Italy, 29 Cliniques Universitaires Saint-Luc, Brussels, Belgium, 30 Centro Hospitalar Universitario de Sao Joao, Porto, Portugal, 31 Department of Neurology, Buffalo General Medical Center, Buffalo, United States of America, 32 Hospital Clinic de Barcelona, Barcelona, Spain, 33 Nemocnice Jihlava, Jihlava, Czech Republic, 34 Azienda Ospedaliera di Rilievo Nazionale San Giuseppe Moscati Avellino, Avellino, Italy, 35 Dept. of Rehabilitation, CRFF Mons. Luigi Novarese, Moncrivello, Italy, 36 MS center, UOC Neurologia, ARNAS Garibaldi, Catania, Italy, 37 Bakirkoy Education and Research Hospital for Psychiatric and Neurological Diseases, Istanbul, Turkey, 38 Ospedali Riuniti di Salerno, Salerno, Italy, 39 Razi Hospital, Manouba, Tunisia, 40 Hospital Universitario Donostia, San Sebastián, Spain, 41 Hospital de Galdakao-Usansolo, Galdakao, Spain, 42 Universitary Hospital Ghent, Ghent, Belgium, 43 The Alfred Hospital, Melbourne, Australia, 44 St. Michael's Hospital, Toronto, Canada, 45 University Hospital Reina Sofia, Cordoba, Spain, 46 Koc University, School of Medicine, Istanbul, Turkey, 47 College of Medicine & Health Sciences and Sultan Qaboos University Hospital, SQU, Oman, 48 Groene Hart Ziekenhuis, Gouda,

**Data Availability Statement:** The data set used in this study is available upon request to the MSBase principal investigators included in the study.

MSBase operates as a single point of contact to facilitate the data sharing agreements with the individual data custodians. Inquiries should be addressed at info@msbase.org. Data is restricted behind a request to ensure a controlled usage of patients data and to stay inline with specific data ownership requirements. The data processing and training scripts to reproduce all experiments are publicly available at https://gitlab.com/edebrouwer/ms_benchmark.

**Funding:** This study was funded by the Research Foundation Flanders (FWO) and the Flemish government through the Onderzoeksprogramma Artificiele Intelligentie (AI) Vlaanderen program (https://www.flandersairesearch.be/en). This funding was awarded to YM, LB, TD, DD, WW, and BDB and funded EBD, TB, LWB, PD, DI, MS, YM, LB, TD, DD, WW, and BDB. EDB was also concomitantly funded by a FWO-SB fellowship (1S98821N - https://fwo.be). The funders had no role in study design, data collection and analysis, decision to publish, or preparation of the manuscript.

**Competing interests:** The authors declare no competing non-financial interests but the following competing financial interests: - Dana Horakova received speaker honoraria and consulting fees from Biogen, Merck, Teva, Roche, Sanofi Genzyme, and Novartis, as well as support for research activities from Biogen and Czech Minsitry of Education [project Progres Q27/LF1]. - Francesco Patti received speaker honoraria and advisory board fees from Almirall, Bayer, Biogen, Celgene, Merck, Novartis, Roche, Sanofi-Genzyme and TEVA. He received research funding from Biogen, Merck, FISM (Fondazione Italiana Sclerosi Multipla), Reload Onlus Association and University of Catania. - Guillermo Izquierdo received speaking honoraria from Biogen, Novartis, Sanofi, Merck, Roche, Almirall and Teva. - Sara Eichau received speaker honoraria and consultant fees from Biogen Idec, Novartis, Merck, Bayer, Sanofi Genzyme, Roche and Teva. - Marc Girard received consulting fees from Teva Canada Innovation, Biogen, Novartis and Genzyme Sanofi; lecture payments from Teva Canada Innovation, Novartis and EMD. He has also received a research grant from Canadian Institutes of Health Research. - Alessandra Lugaresi has served as a Biogen, Bristol Myers Squibb, Merck Serono, Novartis, Roche, Sanofi/ Genzyme and Teva Advisory Board Member. She received congress and travel/ accommodation expense compensations or speaker honoraria from Biogen, Merck, Mylan, Novartis, Roche, Sanofi/Genzyme, Teva and Fondazione Italiana Sclerosi Multipla (FISM). Her

Netherlands, **49** Universidade Metropolitana de Santos, Santos, Brazil, **50** University of Debrecen, Debrecen, Hungary, **51** Liverpool Hospital, Sydney, Australia, **52** Hospital Fernandez, Capital Federal, Argentina, **53** King Fahad Specialist Hospital-Dammam, Khobar, Saudi Arabia, **54** Royal Hobart Hospital, Hobart, Australia, **55** South Eastern HSC Trust, Belfast, United Kingdom, **56** Geneva University Hospital, Geneva, Switzerland, **57** Jahn Ferenc Teaching Hospital, Budapest, Hungary, **58** St Vincent's University Hospital, Dublin, Ireland, **59** University of Western Australia, Nedlands, Australia, **60** Hospital General Universitario de Alicante, Alicante, Spain, **61** Emergency Clinical County Hospital Pius Brinzeu, Timisoara, Romania and University of Medicine and Pharmacy Victor Babes, Timisoara, Romania, **62** Semmelweis University Budapest, Budapest, Hungary, **63** Concord Repatriation General Hospital, Sydney, Australia, **64** AZ Alma Ziekenhuis, Sijsele - Damme, Belgium, **65** Royal Victoria Hospital, Belfast, United Kingdom, **66** AHEPA University Hospital, Thessaloniki, Greece, **67** BAZ County Hospital, Miskolc, Hungary, **68** Mater Dei Hospital, Msida, Malta

☯ These authors contributed equally to this work.
\* liesbet.peeters@uhasselt.be

# Abstract

## Background

Disability progression is a key milestone in the disease evolution of people with multiple sclerosis (PwMS). Prediction models of the probability of disability progression have not yet reached the level of trust needed to be adopted in the clinic. A common benchmark to assess model development in multiple sclerosis is also currently lacking.

## Methods

Data of adult PwMS with a follow-up of at least three years from 146 MS centers, spread over 40 countries and collected by the MSBase consortium was used. With basic inclusion criteria for quality requirements, it represents a total of 15, 240 PwMS. External validation was performed and repeated five times to assess the significance of the results. Transparent Reporting for Individual Prognosis Or Diagnosis (TRIPOD) guidelines were followed. Confirmed disability progression after two years was predicted, with a confirmation window of six months. Only routinely collected variables were used such as the expanded disability status scale, treatment, relapse information, and MS course. To learn the probability of disability progression, state-of-the-art machine learning models were investigated. The discrimination performance of the models is evaluated with the area under the receiver operator curve (ROC-AUC) and under the precision recall curve (AUC-PR), and their calibration via the Brier score and the expected calibration error. All our preprocessing and model code are available at https://gitlab.com/edebrouwer/ms_benchmark, making this task an ideal benchmark for predicting disability progression in MS.

## Findings

Machine learning models achieved a ROC-AUC of 0·71 ± 0·01, an AUC-PR of 0·26 ± 0·02, a Brier score of 0·1 ± 0·01 and an expected calibration error of 0·07 ± 0·04. The history of disability progression was identified as being more predictive for future disability progression than the treatment or relapses history.

institutions received research grants from Novartis and Sanofi Genzyme. - Pierre Grammond has served in advisory boards for Novartis, EMD Serono, Roche, Biogen idec, Sanofi Genzyme, Pendopharm and has received grant support from Genzyme and Roche, has received research grants for his institution from Biogen idec, Sanofi Genzyme, EMD Serono. - Tomas Kalincik served on scientific advisory boards for BMS, Roche, Janssen, Sanofi Genzyme, Novartis, Merck and Biogen, steering committee for Brain Atrophy Initiative by Sanofi Genzyme, received conference travel support and/or speaker honoraria from WebMD Global, Eisai, Novartis, Biogen, Sanofi-Genzyme, Teva, BioCSL and Merck and received research or educational event support from Biogen, Novartis, Genzyme, Roche, Celgene and Merck. - Raed Alroughani received honoraria as a speaker and for serving on scientific advisory boards from Bayer, Biogen, GSK, Merck, Novartis, Roche and Sanofi-Genzyme. - Francois Grand'Maison received honoraria or research funding from Biogen, Genzyme, Novartis, Teva Neurosciences, Mitsubishi and ONO Pharmaceuticals. - Murat Terzi received travel grants from Novartis, Bayer-Schering, Merck and Teva; has participated in clinical trials by Sanofi Aventis, Roche and Novartis. - Jeannette Lechner-Scott travel compensation from Novartis, Biogen, Roche and Merck. Her institution receives the honoraria for talks and advisory board commitment as well as research grants from Biogen, Merck, Roche, TEVA and Novartis. - Samia J. Khoury received compensation for participation in the Novartis Maestro program. - Vincent van Pesch received travel grants from Merck, Biogen, Sanofi, Bristol Myers Squibb, Almirall and Roche; his institution receives honoraria for consultancy and lectures and research grants from Roche, Biogen, Sanofi, Merck, Bristol Myers Squibb, Janssen, Almirall and Novartis Pharma. - Radek Ampapa received conference travel support from Novartis, Teva, Biogen, Bayer and Merck and has participated in a clinical trials by Biogen, Novartis, Teva and Actelion. - Daniele Spitaleri received honoraria as a consultant on scientific advisory boards by Bayer-Schering, Novartis and Sanofi-Aventis and compensation for travel from Novartis, Biogen, Sanofi Aventis, Teva and Merck. - Claudio Solaro served on scientific advisory boards for Merck, Genzyme, Almirall, and Biogen; received honoraria and travel grants from Sanofi Aventis, Novartis, Biogen, Merck, Genzyme and Teva. - Davide Maimone served on scientific advisory boards for Bayer, Biogen, Merck, Sanofi-Genzyme, Novartis, Roche, and Almirall; received honoraria and travel grants from Sanofi Genzyme, Novartis,

## Conclusions

Good discrimination and calibration performance on an external validation set is achieved, using only routinely collected variables. This suggests machine-learning models can reliably inform clinicians about the future occurrence of progression and are mature for a clinical impact study.

### Author summary

Models that accurately predict disability progression in individuals with multiple sclerosis (MS) have the potential to greatly benefit both patients and medical professionals. By aiding in life planning and treatment decision-making, these predictive models can enhance the overall quality of care for people with MS. While previous academic literature has demonstrated the feasibility of predicting disability progression, recent systematic reviews have shed light on several methodological limitations within the existing research. These reviews have highlighted concerns such as the absence of probability calibration assessment, potential biases in cohort selection, and insufficient external validation. Furthermore, the datasets examined often include variables that are not routinely collected in clinical settings or readily available for digital analysis. Consequently, it remains uncertain whether the models identified in these systematic reviews can be effectively implemented in a clinical context. Compounding this issue, the lack of availability of data and analysis code makes it challenging to compare results across different publications. To address these gaps, this study endeavors to develop and validate a machine-learning-based prediction model using the largest longitudinal patient cohort ever assembled for disability progression prediction in MS. Leveraging data from MSBase, a comprehensive international data registry comprising information from multiple MS centers, we aim to create robust models capable of accurately predicting the probability of disability progression. The integration of machine learning models into routine clinical practice has the potential to greatly enhance treatment decision-making and life planning for individuals with MS. The models developed through this study could be subsequently evaluated in a clinical impact study involving MS centers participating in MSBase. This research represents a significant advancement towards the practical application of machine learning models in improving the treatment and care of individuals with MS.

## Introduction

Multiple sclerosis (MS) is a chronic autoimmune disease of the central nervous system [1]. A recent census estimated more than 2·8 million people are currently living with MS [2], which causes a wide variety of symptoms such as mobility problems, cognitive impairment, pain, and fatigue. Importantly, the rate of disability progression is highly variable among people with MS (PwMS) [3]. This heterogeneity makes the personalization of care difficult and prognostic models are thus of high relevance for medical professionals, as they could contribute to better individualized treatment decisions. Indeed, a more aggressive treatment could be prescribed in case of a negative prognosis. Moreover, surveys indicate that PwMS are interested in their prognosis [4], which could help them with planning their lives.

Biogen, Merck, and Roche. - Gerardo Iuliano (retired - no PI successor but has approved ongoing use of data) had travel/accommodations/ meeting expenses funded by Bayer Schering, Biogen, Merck, Novartis, Sanofi Aventis, and Teva. - Bart Van Wijmeersch received research and travel grants, honoraria for MS-Expert advisor and Speaker fees from Bayer-Schering, Biogen, Sanofi Genzyme, Merck, Novartis, Roche and Teva. - Tamara Castillo Triviño received speaking/ consulting fees and/or travel funding from Bayer, Biogen, Merck, Novartis, Roche, Sanofi-Genzyme and Teva. - Jose Luis Sanchez-Menoyo accepted travel compensation from Novartis, Merck and Biogen, speaking honoraria from Biogen, Novartis, Sanofi, Merck, Almirall, Bayer and Teva and has participated in clinical trials by Biogen, Merck and Roche - Guy Laureys received travel and/or consultancy compensation from Sanofi-Genzyme, Roche, Teva, Merck, Novartis, Celgene, Biogen. - Anneke van der Walt served on advisory boards and receives unrestricted research grants from Novartis, Biogen, Merck and Roche She has received speaker's honoraria and travel support from Novartis, Roche, and Merck. She receives grant support from the National Health and Medical Research Council of Australia and MS Research Australia. - Jiwon Oh has received research funding from the MS Society of Canada, National MS Society, Brain Canada, Biogen, Roche, EMD Serono (an affiliate of Merck KGaA); and personal compensation for consulting or speaking from Alexion, Biogen, Celgene (BMS), EMD Serono (an affiliate of Merck KGaA), Novartis, Roche, and Sanofi-Genzyme. - Ayse Altintas received speaker honoraria from Merck, Alexion,; received travel and registration grants from Merck, Biogen - Gen Pharma, Roche, Sanofi-Genzyme. - Yara Fragoso received honoraria as a consultant on scientific advisory boards by Novartis, Teva, Roche and Sanofi-Aventis and compensation for travel from Novartis, Biogen, Sanofi Aventis, Teva, Roche and Merck. - Tunde Csepany received speaker honoraria/ conference travel support from Bayer Schering, Biogen, Merck, Novartis, Roche, Sanofi-Aventis and Teva. - Suzanne Hodgkinson received honoraria and consulting fees from Novartis, Bayer Schering and Sanofi, and travel grants from Novartis, Biogen Idec and Bayer Schering. - Norma Deri received funding from Bayer, Merck, Biogen, Genzyme and Novartis. - Bruce Taylor received funding for travel and speaker honoraria from Bayer Schering Pharma, CSL Australia, Biogen and Novartis, and has served on advisory boards for Biogen, Novartis, Roche and CSL Australia. - Fraser Moore participated in clinical trials sponsored by EMD Serono and Novartis. - Orla Gray received

There is a large amount of literature on prognostic MS models [5–10]. Some prognostic models are or were at some point available as web tools. However, with the exception of Tintore et al. [10] that focuses on conversion to MS, none have been integrated into clinical practice and no clinical impact studies have been performed [5, 6]. Because MS is a complex chronic disease that is often treated within a multidisciplinary context, the performance of a prognostic model studied in isolation from its clinical context gives limited information on its clinical relevance [11, 12]. Recent systematic reviews have highlighted several methodological issues within the current literature [5, 6], such as the lack of calibration or a possible significant bias in the cohort selection. Moreover, the investigated datasets are rarely made available. They furthermore often contain variables that are not routinely collected within the current clinical workflow (*e.g.* neurofilament light chain) or are not readily available for digital analysis (*e.g.* Magnetic Resonance Imaging (MRI) images).

In this article, we aimed at developing a model with three specific goals. Firstly, it should predict the *probability* (a value between 0 and 1) of disability progression for a PwMS within the next two years, instead of just a binary target (0 or 1, i.e., disease progression or no disease progression). Secondly, it should be applicable to a well-defined, relevant, and large patient population. Thirdly, all variables used in the model should be available in routine clinical care. A successful combination of these three goals would justify a clinical impact study of the model and represent a significant step towards clinical applicability.

With this aim in mind, we developed and externally validated machine learning models to predict disability progression after two years for PwMS, using commonly-available clinical features. For this task we represented disability progression as a binary variable indicating if a confirmed disability progression will occur within the next two years, as defined by Kalincik et al. [13]. We trained the models using the largest longitudinal patient cohort to date for disability progression prediction in MS. The cohort was extracted from MSBase, a large international data registry containing data from multiple MS centers. We evaluated the performance, including the predicted probabilities, of different machine learning architectures and found they could achieve a ROC-AUC of $0.71 \pm 0.01$, an AUC-PR of $0.26 \pm 0.02$, a Brier score of $0.1 \pm 0.01$, and an expected calibration error of $0.07 \pm 0.04$.

Importantly, and in contrast with the available literature on disease progression models for MS (except for one model to predict relapses [14]), our data pre-processing pipeline and our models check all the boxes of the Transparent Reporting for Individual Prognosis Or Diagnosis (TRIPOD) checklist. Our work therefore provides an important step towards the integration of artificial intelligence (AI) models in MS care. The outline of our approach is presented in Fig 1.

## Results

### Cohort statistics

In this multi-center international study, we used data of people with MS from 146 centers spread over 40 different countries and compiled in the MSBase registry [15] as of September 2020. All data were prospectively collected during routine clinical care predominantly from tertiary MS centres [16].

The inclusion criteria for the initial extraction of the data from MSBase were: having at least 12 months of follow-up, being aged 18 years or older, and diagnosed with relapsing remitting (RR) MS, secondary progressive (SP) MS, primary progressive (PP) MS. Clinically-isolated syndrome (CIS) patients were excluded. This resulted in an initial cohort of 40,827 patients.

honoraria as consultant on scientific advisory boards for Genzyme, Biogen, Merck, Roche and Novartis; has received travel grants from Biogen, Merck, Roche and Novartis; has participated in clinical trials by Biogen and Merck. - Csilla Rozsa received speaker honoraria from Bayer Schering, Novartis and Biogen, congress and travel expense compensations from Biogen, Teva, Merck and Bayer Schering. - Allan Kermode received speaker honoraria and scientific advisory board fees from Bayer, BioCSL, Biogen, Genzyme, Innate Immunotherapeutics, Merck, Novartis, Sanofi, Sanofi-Aventis, and Teva. - Magdolna Simo received speaker honoraria from Novartis, Biogen, Bayer Schering; congress/travel compensation from Teva, Biogen, Merck, Bayer Schering. - Todd Hardy has received speaking fees or received honoraria for serving on advisory boards for Biogen, Merck, Teva, Novartis, Roche, Bristol-Myers Squibb and Sanofi-Genzyme, is Co-Editor of Advances in Clinical Neurosciences and Rehabilitation, and serves on the editorial board of Journal of Neuroimmunology and Frontiers in Neurology. - Nikolaos Grigoriadis received honoraria, consultancy/lecture fees, travel support and research grants from Biogen Idec, Biologix, Novartis, TEVA, Bayer, Merck Serono, Genesis Pharma, Sanofi – Genzyme, ROCHE, Cellgene, ELPEN and research grants from Hellenic Ministry of Development.

The clinical trajectory of each patient in the cohort consisted of multiple, potentially overlapping, clinical *episodes*, that allowed to artificially augment the dataset. We defined a clinical episode as the conjunction of an observation window, a baseline EDSS measurement, and a disability progression label. Details about the construction of the clinical episodes are given in the Materials and Methods. For each episode, we required a minimum of three EDSS measurements over the last three years and three months at the time of prediction. This inclusion criterion represents the typical follow-up frequency for PwMS, which is once or twice a year. Prior work showed that longitudinal clinical history was beneficial for prediction of disability progression [17]. The final cohorts resulted in a total of 283,115 valid episodes from 26,246 patients. Basic characteristics of the final cohort are shown in Table 1.

## Model performance

The performance of the predictive models assessed on the external test cohort is reported in Tables 2–4. A visual illustration of the discrimination performance is shown in Fig 2. A temporal-attention-based model reached an area under the receiver operating characteristic curve (ROC-AUC) of $0·71 \pm 0·01$ and an area under the precision-recall curve (AUC-PR) of $0·26 \pm 0·02$, with a calibration error of $0·07 \pm 0·04$ on the external test cohort.

To assess the reliability of those results on specific subgroups of patients, we also evaluated the performance for each different MS course at the time of prediction (Table 3) and different baseline EDSS ($EDSS_{t=0}$) (Table 4). The relapsing-remitting (RR) category showed a performance similar to the full cohort. We observed a decreased discrimination performance in the progressive and secondary progressive groups. We conjecture that this is due to the low sample size in these groups. A similar effect was observed when segmenting by disability severity, with the group of higher severity showing a lower discrimination performance. In the supporting information, we also present a segmentation of the results by the medical center of origin of the patients (S1, S2 and S3 Figs), indicating a higher variability of the results for small centers.

The calibration of the different models was assessed from the Brier score and the expected calibration errors (ECE), which are reported in Tables 2–4. In Fig 3, we report the calibration plot of the longitudinal attention model on the external test cohort. We observed a very good calibration of the predicted risks in the range between 0 and 0.3, suggesting an excellent reliability of the predictive model. The calibration curves of other models are given in the supporting information (S4 Fig) along with a segmentation of the calibration of the models by clinical subgroups (S5 Fig). A comprehensive comparison of all considered models is available in S2, S3, S4, S5, S6 and S7 Tables.

## Feature importance

The importance of the different variables used in the machine learning models was investigated. Fig 4 shows the results of a permutation importance test on the multi-layer percepetron (MLP) model, by assessing the loss in discrimination performance when a variable is shuffled over the test set [19]. Fig 4 ranks the features in decreasing order of importance. We found the most important variables to be the baseline EDSS at prediction time, the number of years since 1990 as well as the mean EDSS and Kurtzke Functional Systems Score (KFS) over the last 3 years. The complete results are available in S8 Table.

The baseline EDSS was expected to be important in the prediction, as the definition of the progression event directly depends on it (as seen in Eq (1)). The time since 1990 suggests a change of behavior of the disease over the years that could be explained by progress in clinical care or enhanced diagnosis of earlier and milder forms of the disease. The importance of the previous values of EDSS and KFS demonstrates an added value of considering longitudinal

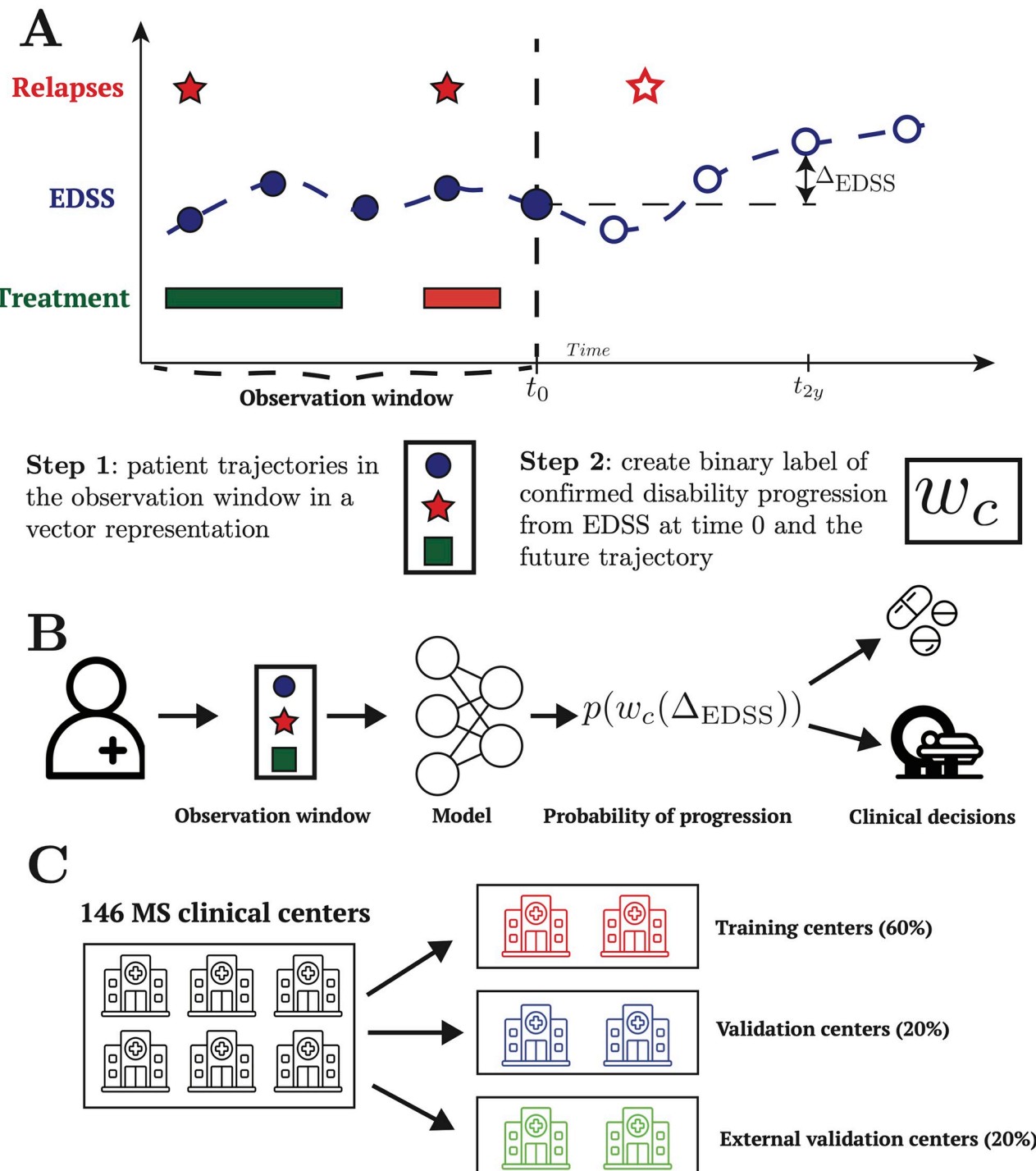

**Fig 1. Overall layout of our approach.** A: Representation of a clinical trajectory of an individual person with multiple sclerosis (PwMS). The trajectory consists of, among others, relapses, EDSS values, and treatment durations collected over time. The full list of used variables is given in the Materials and Methods. The trajectory of each patient is divided into an observation window (the available clinical history for the prediction) and the future trajectory, which is used to compute the confirmed disability progression label at two years ($w_c$). B: For an individual PwMS, the clinical trajectory in the observation window is extracted and used in the machine learning model to predict a well-calibrated probability of disability progression at two years. Based on the predictions, clinicians can adjust their clinical decisions accordingly. C: The MSbase dataset contains clinical data from 146 individual MS clinical centers with different clinical practice. We leveraged this feature by creating an external validation cohort of patients. We split the data per clinic, with 60% of patients used for training the model, 20% for optimizing the hyper-parameters (validation set) and 20% for external validation. The results presented in this work are all on the external validation cohort.

**Table 1. Summary statistics of the cohort of interest after extraction from MSBase (Extracted Cohort) and after patient and sample selection (Final Cohort).** For all variables the value at the last recorded visit was used. KFS stands for Kurtzke Functional Systems Score, DMT for Disease Modifying Therapy, CIS for Clinically Isolated Syndrome.

| Variable | Cohort 3 EDSS |
| --- | --- |
| Patients (% female) | 26,246 (71·8) |
| Age, Years[a] | 42·8 (10·8) |
| Age at MS onset, years[a] | 31·3 (8·9) |
| Disease duration, years[a] | 11·6 (8·0) |
| Education status, % higher[c] | 18·2 (65·1) |
| First symptom, none given (%) | 13·7 |
| supratentorial (%) | 28·2 |
| optic pathways (%) | 22·6 |
| brainstem (%) | 24·3 |
| spinal cord (%) | 26·4 |
| MS course | / |
| CIS (%) | 0 |
| Relapsing-Remitting (%) | 83·5 |
| Primary Progressive (%) | 5·0 |
| Secondary Progressive (%) | 11·5 |
| EDSS[a] | 3·0 (2·1) |
| $EDSS_{t=0}$ category | / |
| $EDSS_{t=0} \leq 5\cdot5$ (%) | 83·9 |
| $EDSS_{t=0} > 5\cdot5$ (%) | 16·1 |
| Annualized relapse rate[b] | 0·82 [0·43, 1·47] |
| KFS Scores | / |
| pyramidal[b] | 2 [1, 3] |
| cerebellar[b] | 0 [0, 2] |
| brainstem[b] | 0 [0, 1] |
| sensory[b] | 1 [0, 2] |
| sphincteric[b] | 0 [0, 1] |
| visual[b] | 0 [0, 1] |
| cerebral[b] | 0 [0, 1] |
| ambulatory[b] | 0 [0, 1] |
| DMT | / |
| none | 23·5 |
| low-efficacy | 51·3 |
| moderate-efficacy | 13·6 |
| high-efficacy | 11·6 |
| high induction | 7·2 |

[a]: mean ± standard deviation

[b]: median (quartiles)

[c]: % missing data

data, as already shown in De Brouwer et al. [17]. Remarkably, no variables including disease modifying treatments (DMT) were given a significant importance score.

## Discussion

The models investigated in this study provide a significant advance towards deploying AI in clinical practice in MS. After validation of the results in a clinical impact study, they have the

**Table 2. Summary statistics of the performance measures (averages ± standard deviations).** Baseline performance are 0·5 for the area under the receiver operating curve (ROC-AUC) and 0·11 for the area under the precision-recall curve (AUC-PR). ↑ indicates higher is better. ↓ indicates lower is better. p-value for ROC-AUC between Ensemble and MLP: 0·152 for unpaired t-test. p-value for AUC-PR between Attention and MLP: 0.452 for unpaired t-test.

| Model | ROC-AUC ↑ | AUC-PR ↑ | Brier ↓ | ECE ↓ |
|---|---|---|---|---|
| Ensemble | 0·71 ± 0·01 | 0·25 ± 0·02 | 0·10 ± 0·01 | 0·06 ± 0·05 |
| Attention | 0·71 ± 0·01 | 0·26 ± 0·02 | 0·10 ± 0·01 | 0·07 ± 0·04 |
| Bayesian NN | 0·71 ± 0·01 | 0·25 ± 0·01 | 0·10 ± 0·01 | 0·08 ± 0·04 |
| MLP | 0·70 ± 0·01 | 0·24 ± 0·02 | 0·10 ± 0·01 | 0·09 ± 0·03 |

**Table 3. Results for disability progression prediction per MSCourse (Primary Progressive (PP), Relapsing Remitting (RR), and Secondary Progressive (SP)), for the best models.** ↑ indicates higher is better. ↓ indicates lower is better. We report averages ± standard deviations computed over the different folds. Training sizes of different groups: PP = 10,976 episodes (1,192 patients); RR = 185,724 episodes (16,268 patients); SP = 23,704 episodes (2,402 patients).

| Model | MSCourse | ROC-AUC ↑ | AUC-PR ↑ | Brier ↓ | ECE ↓ |
|---|---|---|---|---|---|
| Attention | PP | 0·65 ± 0·01 | 0·33 ± 0·04 | 0·16 ± 0·01 | 0·07 ± 0·02 |
| Attention | RR | 0·70 ± 0·01 | 0·21 ± 0·01 | 0·09 ± 0·01 | 0·06 ± 0·03 |
| Attention | SP | 0·65 ± 0·01 | 0·33 ± 0·03 | 0·17 ± 0·01 | 0·10 ± 0·05 |
| Bayesian NN | PP | 0·66 ± 0·01 | 0·34 ± 0·03 | 0·16 ± 0·01 | 0·09 ± 0·05 |
| Bayesian NN | RR | 0·70 ± 0·01 | 0·20 ± 0·01 | 0·09 ± 0·01 | 0·09 ± 0·03 |
| Bayesian NN | SP | 0·64 ± 0·01 | 0·32 ± 0·02 | 0·17 ± 0·01 | 0·11 ± 0·02 |
| MLP | PP | 0·63 ± 0·03 | 0·32 ± 0·05 | 0·16 ± 0·01 | 0·09 ± 0·03 |
| MLP | RR | 0·69 ± 0·01 | 0·19 ± 0·0 | 0·09 ± 0·01 | 0·05 ± 0·02 |
| MLP | SP | 0·63 ± 0·01 | 0·31 ± 0·02 | 0·17 ± 0·01 | 0·10 ± 0·04 |

**Table 4. Results for disability progression prediction for different baseline Expanded Disability Status Scale score ($EDSS_{t=0}$), $EDSS_{t=0} \leq 5\cdot5$ and $>5\cdot5$.** ↑ indicates higher is better. ↓ indicates lower is better. We report averages ± standard deviations computed over the different folds. Traning size of the different groups: $EDSS_{t=0} \leq 5\cdot5$ = 185,556 episodes (16,282 patients); $EDSS_{t=0} > 5\cdot5$ = 34,848 episodes (4,686 patients).

| Model | $EDSS_{t=0}$ | ROC-AUC ↑ | AUC-PR ↑ | Brier ↓ | ECE ↓ |
|---|---|---|---|---|---|
| Attention | $EDSS_{t=0} \leq 5\cdot5$ | 0·72 ± 0·01 | 0·26 ± 0·01 | 0·09 ± 0·0 | 0·07 ± 0·04 |
| Attention | $EDSS_{t=0} > 5\cdot5$ | 0·65 ± 0·01 | 0·27 ± 0·04 | 0·15 ± 0·01 | 0·07 ± 0·02 |
| Bayesian NN | $EDSS_{t=0} \leq 5\cdot5$ | 0·72 ± 0·01 | 0·25 ± 0·01 | 0·09 ± 0·0 | 0·08 ± 0·04 |
| Bayesian NN | $EDSS_{t=0} > 5\cdot5$ | 0·64 ± 0·02 | 0·26 ± 0·03 | 0·15 ± 0·01 | 0·11 ± 0·03 |
| MLP | $EDSS_{t=0} \leq 5\cdot5$ | 0·71 ± 0·01 | 0·24 ± 0·01 | 0·1 ± 0·01 | 0·09 ± 0·03 |
| MLP | $EDSS_{t=0} > 5\cdot5$ | 0·63 ± 0·01 | 0·26 ± 0·04 | 0·15 ± 0·02 | 0·09 ± 0·03 |

potential to let the research in MS benefit from the advantages of advanced predictive modeling capabilities.

Our work confirms that predicting the probability of disability progression of MS patients is feasible. Importantly, despite MS progression being inherently stochastic, this study shows that relevant historical clinical data, collected as part of routine clinical care, can lead to high discrimination performance and good calibration (Fig 3), which is crucial in healthcare applications. Combined with our rigorous benchmarking, external validation, and our strict adherence to the TRIPOD guidelines, this points towards a readiness of these models to be tested in a clinical impact study. Such study would evaluate the performance of these models in real-world clinical practice, compare them with the predictions of clinicians, and assess the value of such a prediction for PwMS. Over- or under-prediction of the probability of progression could indeed lead to unnecessary emotional stress or optimism.

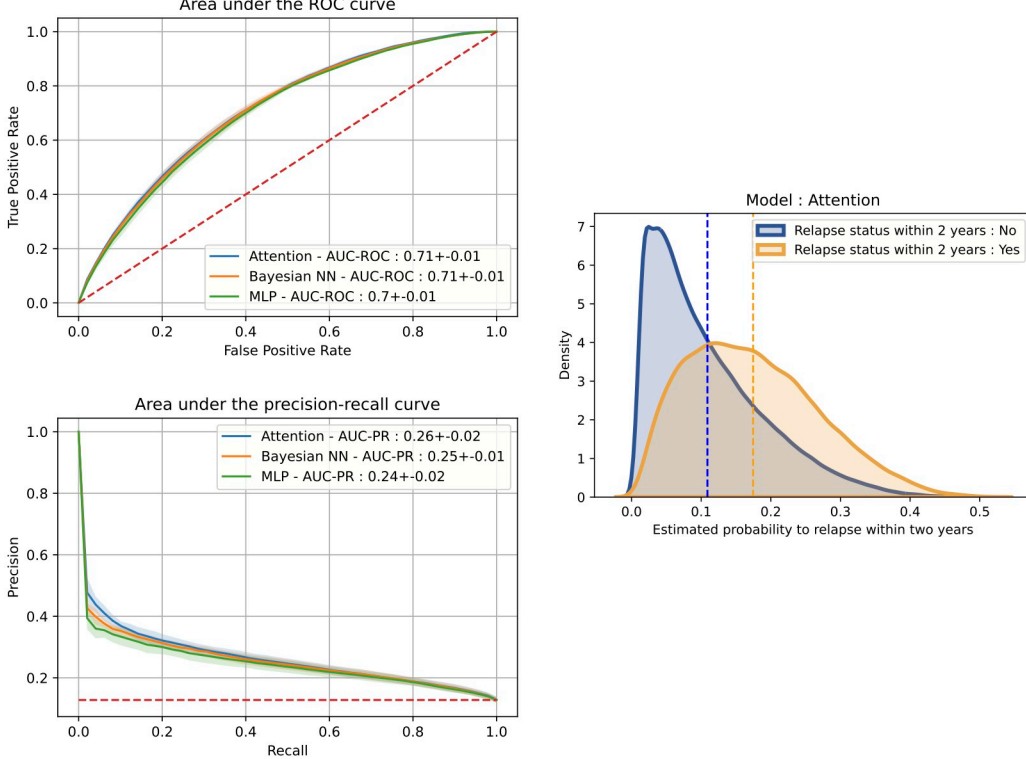

**Fig 2. Visual representation of the discrimination performance.** ROC-AUC curve, the AUC-PR curve, and distribution of the estimated probability of relapse per group obtained with the temporal attention model.

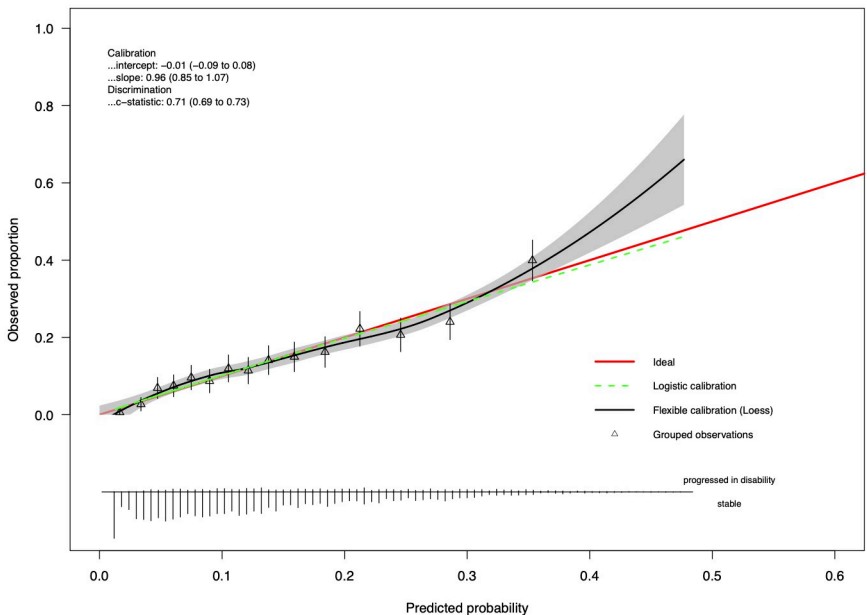

**Fig 3. Calibration diagram for the temporal attention model for the first data split.** The *val.prob.ci.2* function [18] was used to generate this plot.

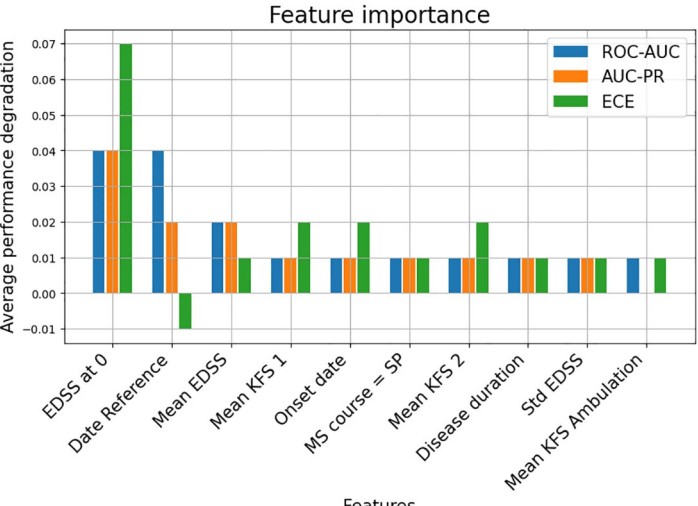

**Fig 4. Feature importance of different variables.** Feature importance of different variables used in the MLP model based on the average performance degradation on the ROC-AUC, AUC-PR, and ECE metrics. 'EDSS at 0' stands for the Expanded Disability Status Scale score at the time of prediction. 'Date Reference' represents the date of prediction. 'Mean EDSS' stands for the average EDSS over the last 3 years. 'MS Course = SP' is a binary variable indicating that the MS course is secondary progressive at the time of prediction. 'Mean KFS x' represents the corresponding variable in the average Kurtzke Functional Systems Score over the last three years. 'Std EDSS' represents the standard deviation of EDSS over the last 3 years.

Our attained ROC-AUC scores of 0.71 are compatible with those found in the literature, which were found to range between 0.64 and 0.89 [5]. Our ROC-AUC scores are on the low end of this range. This could be explained by several factors, such as: MSBase being a large and diverse population; the use of a limited set of variables, since we constrained ourselves to variables that are collected during routine clinical care; a validation set-up where prediction is done on patients from different clinics than those in the training set.

Previous work had only reported calibration graphically [20–22], with some of these models showing good calibration. The possibility to achieve well-calibrated models is empirically confirmed in our study. As previous studies used different patient populations, covariates and prediction targets, we could not directly compare our models with other models from previous studies.

The models developed in this study also suffer from limitations. First of all, several countries with good quality MS registries were not included because they are not part of the MSBase initiative. Since treatment decisions can be country specific to a significant degree [23], it can result in a difference of performance of the proposed models on countries not included in this dataset. Yet, a clinical impact study in MS centers participating in MSBase would not suffer from such external validity problems.

Second, our inclusion criteria required patients with good follow-up (at least one yearly visit with EDSS measurement), so stable patients that do not visit regularly, or newly diagnosed with MS cannot benefit from these models. This limits the application to patients with an already established clinical MS history. This decision was motivated by prior work [17], which showed that including clinical history as a predictor leads to more accurate prognosis, a finding that we confirm in this study. A new dedicated model would be required for disability progression in patient with shorter clinical history. Nevertheless, MS being a chronic disease, many patients would still satisfy our follow-up inclusion criteria (64% in the MSBase cohort).

Third, our analysis showed that the performance of the different models varied across different patient subgroups. When segmenting the cohort by disease course or by baseline EDSS,

we found that the majority subgroup (*i.e.* relapsing-remitting and $EDSS_{t=0} \leq 5\cdot5$) showed a better discrimination performance than subgroups with lower prevalence. We conjecture that this difference of performance is due to the lower sample size in the minority subgroups. This finding suggests a more limited value of the models for PwMS belonging to the minority subgroups. Nevertheless, the difference in calibration was not significantly different.

Fourth, the progression target that we defined in this work cannot realistically fully capture the complexity of the disease and progression in MS cannot be summarized by EDSS only. EDSS itself, as an attempt to quantify progression on one-dimensional scale, lacks the expressivity to reliably encode the progression of the disease. What is more, we framed disability progression as a classification task, which is more granular than predicting future EDSS, but is more amenable for machine learning. Despite these imperfections, the confirmed disability progression label used in this work has been proven clinically useful [13], striking a good balance between abstraction and expressivity. Our work builds upon those concepts and inherits their flaws and advantages.

Despite these imperfections, our models could potentially help patients in the planning of their lives and provide a baseline for further research. An emphasis on reproducibility was made, in an attempt to provide a strong benchmark for this important task. Thanks to the excellent clinically-informed pre-processing pipeline, researchers can easily extend the current models or propose their own, to continuously improve disease progression prediction. Extensions to our method could include treatment recommendation or inclusion of other biomarkers available in a specific center.

## Materials and methods

### Resource availability

**Lead contact.** Further information and requests for resources should be directed to and will be fulfilled by the lead contact, Liesbet Peeters (liesbet.peeters@uhasselt.be).

**Materials availability.** Trained machine learning models can be found at https://gitlab.com/edebrouwer/ms_benchmark.

### Cohort definition

In this multi-center international study, we used data of people with MS from 146 centers spread over 40 different countries and compiled in the MSBase registry [15] as of September 2020. All data were prospectively collected during routine clinical care predominantly from tertiary MS centres [16].

The inclusion criteria for the initial extraction of the data from MSBase were: at least 12 months of follow-up, aged 18 years or older, and diagnosed with relapsing remitting (RR) MS, secondary progressive (SP) MS, primary progressive (PP) MS. Clinically-isolated syndrome (CIS) patients were excluded from the study. This initial dataset contained a total of 44,886 patients.

In order to ensure data quality, some observations or patients were removed from that cohort. Exclusion criteria include:

- Visits of the same patient that happened on the same day but had different expanded disability status scale (EDSS) values were removed. All duplicate visits with the same EDSS for the same visit date were removed (i.e., only one of the visits was retained). Visits from before 1970 were discarded.

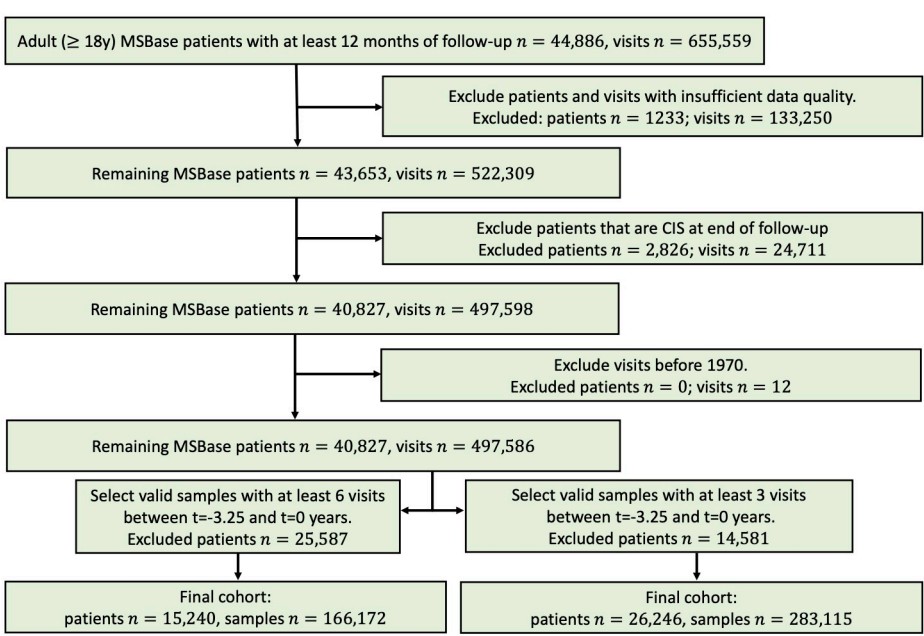

**Fig 5. Flowchart of patient selection.** Flowchart of patient selection for both at least three and at least six visits in the last 3.25 years.

- Patients with the CIS MS course at their last visit were discarded. For those patients the relevant question is whether or not they will progress to confirmed MS, which is a different question than the one investigated in this work.

- Patients whose diagnosis date or age at first symptoms (i.e., MS onset date) was missing or with invalid formatting were removed.

- Patient whose MS course or sex was not available were removed.

- Patients whose date of MS diagnosis, birth, MS onset, start of progression, clinic entry or first relapse was higher than the extraction date were discarded.

- All visits whose visit date had an invalid format or was after the extraction date were discarded.

These criteria resulted in a total number of 40, 827 patients in the cohort. A flowchart of the patient inclusion for the final cohort is shown in Fig 5. Basic characteristics of the final cohorts are shown in Table 1.

The clinical trajectory of each patient in the cohort consisted of multiple, potentially overlapping, clinical episodes, that allow to artificially augment the dataset. Clinical episodes are defined in the sections below.

External validation was used to assess the performance of our predictive models by splitting the cohort by MS center. The models were thus evaluated on patients from different clinics than the ones used for training. An assessment of the heterogeneity across centers is available in the supporting information (S3 Fig).

## Definition of disability progression

Machine learning models were trained to predict a disability progression binary variable for each clinical episode. In this section, we describe the definition of this binary disability

progression label. Conceptually, disability progression is defined as a sustained increase in EDSS over time.

Because assessing progression requires a baseline EDSS value to compare with, predictions were made at visit dates where an EDSS measurement was recorded. In our notation, $t = 0$ denotes the time of the visits at which the prediction is made and the baseline EDSS is thus written as $\text{EDSS}_{t=0}$. Motivated by the non-linearity of the EDSS, unconfirmed disability progression ($w = 1$) after two years ($t = 2y$) is defined as follows [13]:

$$
w \quad = \quad
\begin{cases}
1 & \text{if } \text{EDSS}_{t=2y} - \text{EDSS}_{t=0} \geq 1 \cdot 5 \ \& \ \text{EDSS}_{t=0} = 0 \\
1 & \text{if } \text{EDSS}_{t=2y} - \text{EDSS}_{t=0} \geq 1 \ \& \ 0 < \text{EDSS}_{t=0} \leq 5 \cdot 5 \\
1 & \text{if } \text{EDSS}_{t=2y} - \text{EDSS}_{t=0} \geq 0 \cdot 5 \ \& \ \text{EDSS}_{t=0} > 5 \cdot 5 \\
0 & \text{otherwise.}
\end{cases}
\tag{1}
$$

$\text{EDSS}_{t = 2y}$ represents the last recorded EDSS before $t = 2$ years. We chose a time horizon of two years as a trade-off between short and long disease time scales. A short horizon would lead to very few confirmed progression in the cohort, making predictive modeling difficult. A long horizon would result in less patients satisfying the inclusion criteria, reducing the sample size. It is a typical choice in the literature and is a relevant timescale for PwMS to plan their lives.

EDSS suffers from inter- and intra-rater variability [24]. The actual state of the patient also fluctuates, because of e.g. recent relapses from which the patient could still (partly) recover. We therefore studied *confirmed* disability progression ($w_c$) for at least six months. Progression was confirmed if all EDSS values measured within six months after the progression event and the first EDSS measurement after two years lead to the same worsening target $w = 1$ according to Eq (1). EDSS measurements within one month after a relapse were not taken into account for confirming disability progression [13]. $w_c$ represents the target binary label used to train the machine learning models.

Importantly, if progression ($w = 1$) could be confirmed because there were no EDSS measurements after two years that could be used for confirmation, it was not considered a valid target and no label would be derived. If progression could not be confirmed because an EDSS used for confirmation led to $w = 0$, it counted as no confirmed disability progression ($w_c = 0$). If there was no disability progression ($w = 0$), no confirmation was needed to make it a valid target. Note that even with confirmation for at least six months, around 20% of progression events were expected to regress after more than five years [25]. However, disability progression that lasts several years is a relevant outcome for a person with MS.

We note that the above definition of confirmed disability progression has been introduced and clinical motivated in Kalincik et al. [13]. Although it is only a surrogate for the actual and complex disease progression mechanism, it represents a clinically validated label that is more amenable to statistical and machine learning analysis.

### Definition of clinical episodes

For each patient, all visits can potentially represent a valid baseline EDSS for a progression episode. More generally, it is possible to divide the available clinical history of a patient in multiple (potentially overlapping) episodes for which a disability progression label can be computed. Each episode therefore consists of an observation window (the clinical history before $t = 0$), a baseline EDSS ($\text{EDSS}_{t=0}$) and a confirmation label ($w_c$) as shown on Fig 6. Extracting several episodes per patient allowed to significantly increase the number of data points in the study.

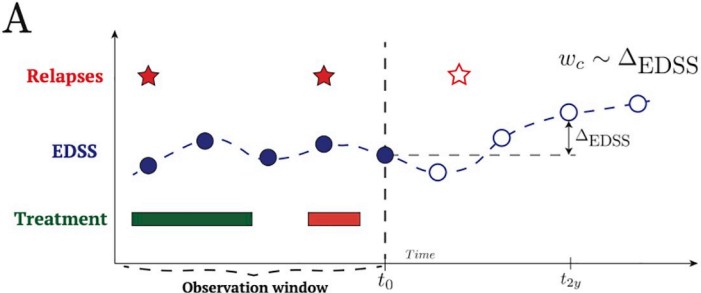

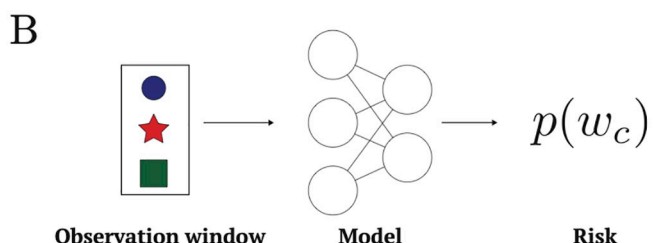

**Fig 6. Problem Setup.** A: For each patient episode, the available data for prediction consists of the baseline data and the longitudinal clinical data in the observation window. Disability progression ($w_c$) was assessed based on the difference between the EDSS at time $t = 0$ and two years later ($t = t_{2y}$) as defined in Eq (1). B: Based on the available historical clinical data (in the observation time window), we aimed at training a model able to predict the probability $p$ ($w_c$) of disability progression at a two years horizon ($t_{2y}$).

To assess the impact of follow-up on the performance of the models, we defined two cohorts of patients, one with a minimum of three EDSS measurements, the other with a minimum of six EDSS measurements over the last three years and three months of the observation window. While our results focus on the cohort with a minimum of three EDSS measurements, performance results for the other cohort are presented in the supporting information. The three measurements requirement excluded patients who have a less than yearly (or biyearly) EDSS follow-up frequency. The three additional months were chosen to allow for some margin regarding when the yearly visit was planned.

Episodes were considered valid if they met the following criteria:

- A valid confirmed disability progression label ($w_c$) could be computed at $t = 0$.

- The time at which the prediction were made was after 1990 ($t_0 > 1990$, Jan 1st). This ensured that we had a cohort of patients from decades were disease modifying therapies (DMTs) were available [26].

- There were at least $k$ EDSS measurement in the last last three years and three months of the observation window, where $k$ is either three or six measurements.

Examples of valid and invalid episodes are presented in Fig 7. The final cohorts resulted in a total of 283,115 valid episodes from 26,246 patients, for a minimum of three EDSS measurements and 166,172 valid episodes from 15,240 patients, for a minimum of six EDSS measurements. For the 3-visits cohort, 11·64% of the episodes represented a progression event, hence showing a mild imbalance. We addressed this imbalance by re-weighing each sample proportionally to its label occurrence.

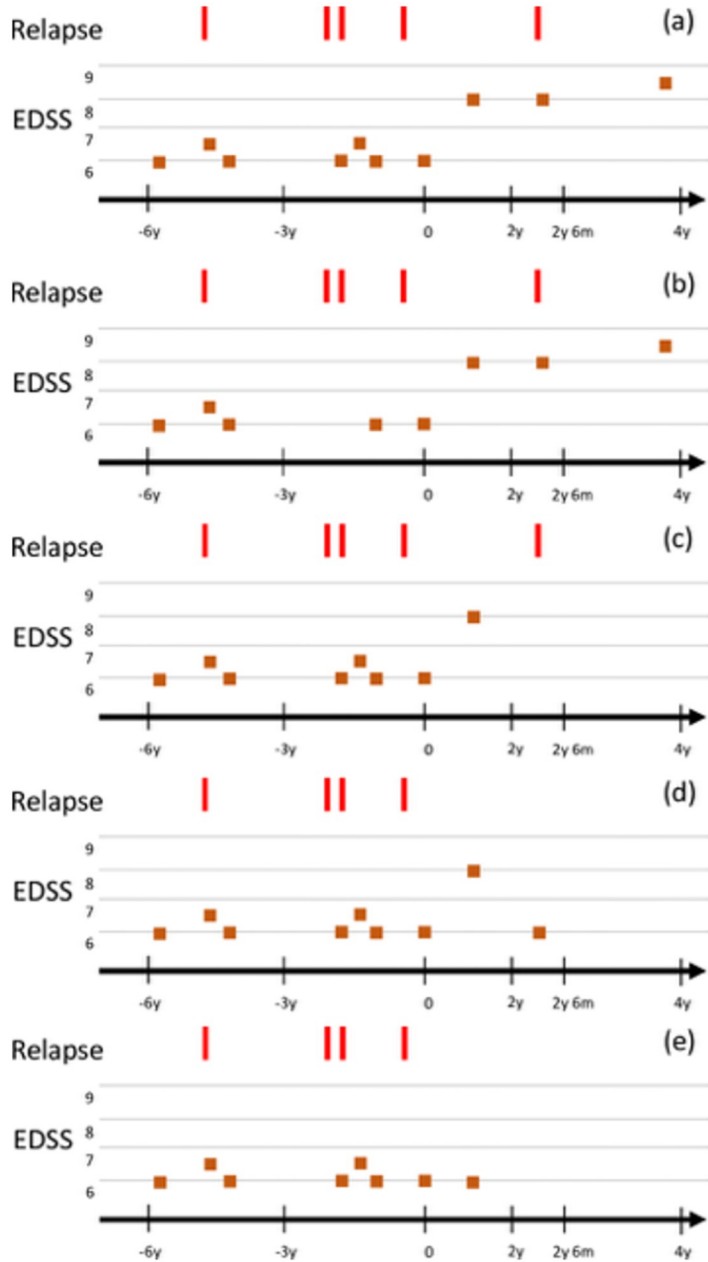

**Fig 7. Examples of valid and non-valid episodes.** The time is in years ($y$) and months ($m$). (a) Confirmed progression after two years. The EDSS around $2y6m$ is not used to confirm the progression, because it occurs within 1 month after a relapse. Progression is confirmed with the EDSS measurement around $4y$. There are 3 EDSS measurements between $-3y$ and $0y$, which is enough follow-up data. (b) This is not a valid sample: there are not enough EDSS measurements between $-3y$ and $0y$. (c) This is not a valid sample: no confirmed progression because there are no EDSS values after $2y$. (d) This is a valid sample: the EDSS decreases after $2y$, so this counts as no disability progression. (e) This is a valid sample: $w_u = 0$, so no confirmation is needed.

## Variables

A set of clinical variables was retained from all available variables and included in the observation window of each episode. The following static (*i.e.*, non-varying over time) variables were selected: birth date, sex, MS onset date, education status (higher education, no higher

education, unknown) and the location of the first symptom (*i.e.*, supratentorial, optic pathways, brainstem or spinal cord).

The following longitudinal variables were also collected in the observation window (*i.e.*, for times $t \leq 0$): EDSS, MS course (Relapsing Remitting MS (RRMS), Primary Progressive MS (PPMS), Secondary Progressive MS (SPMPS), Clinically Isolated Syndrome (CIS)), relapse occurrence, relapse position (pyramidal tract, brainstem, bowel bladder, cerebellum, visual function, sensory), all Kurtzke functional system (KFS) scores, and Fampridine administration. The disease modifying therapies (DMT) and immunosuppressants were categorized into low-efficacy, moderate-efficacy and high-efficacy:

- Low-efficacy: Interferons, Teriflunomide, Glatiramer, Azathioprine, Methotrexate.

- Moderate-efficacy: Fingolimod, Dimethyl-Fumarate, Cladribine, Siponimod, Daclizumab

- High-efficacy: Alemtuzumab, Rituximab, Ocrelizumab, Natalizumab, Mitoxantrone, Cyclophosphamide.

Except for Mitoxantrone and Cyclophosphamide, we assumed that only one DMT was administered at the same time. This implies that if a new DMT was started, the administration of the previous DMT was considered to have ended, even if no end date was registered in the data. Mitoxantrone and Cyclophosphamide can be administered in combination with another DMT. Indeed, they are induction DMTs and are thus expected to have a long-term effect. Therefore, only the start dates of these two DMTs were recorded. They were coded by a separate category: highly active induction DMTs. Alemtuzumab and Cladribine are also induction DMTs. In contrast to Mitoxantrone and Cyclophosphamide they are not combined with other DMTs. If a new DMT was started, it was assumed that they were considered as not effective and the start date of the new DMT was taken as the end date of Alemtuzumab or Cladribine.

MRI variables were not included due to high missingness. Indeed, the lesion counts were available in less than 1·7% of the clinical episodes. The variable indicating whether the MRI was normal, abnormal MS typical, or abnormal MS atypical was judged as not informative enough.

The above variables were then grouped in three feature sets: *static*, *dynamic* (summary statistics of the clinical history) and *longitudinal* [17]. These represent increasing quantity of information regarding the clinical history of patients.

**Grouping of the included clinical variables.**   The *static feature set* contains variables available at $t = 0$ without taking into account possible previous values. Categorical variables can be encoded as indicator variables. For example, sex is encoded as female 'yes / no' and male 'yes / no'. If that feature contains missing occurrences, the category 'unknown' is added. EDSS and the KFS scores were treated as continuous variables, even though they are categorical. The variables of the static feature set are: Sex, Age (years), Age at MS onset (years), Disease duration (years), MS course at $t = 0$ (RRMS, SPMS, PPMS), EDSS at $t = 0$, Last used DMT at $t = 0$, Use of induction DMT at $t = 0$, all KFS scores at $t = 0$, education status, first symptom (supratentorial, optic pathways, brainstem, spinal cord or missing), time of prediction (years since 1990), and time of diagnosis (years since 1990).

The *dynamic feature set* adds information about the clinical history before $t = 0$ (longitudinal information) to the static dataset. It contains variables that are hand-engineered from the longitudinal variables: number of visits in the last 3.25 years, the minimum and maximum in the whole history ($t \leq 0$) of the EDSS and all KFS variables, mean and standard deviation over the last 3.25 years of the EDSS and all KFS scores, oldest EDSS and KFS score measured in the last 3.25 years, relapse rate over the whole history (number of relapses divided by the follow-up period—since first clinical visit), time since the last relapse (years), presence of high-efficacy DMT in the past, disease duration until a first DMT was administered, disease duration until

an high-efficacy active DMT was administrated, time spent on a DMT during the disease duration (ratio of time on a DMT divided by the time since MS onset), and time since the last Fampridine administration.

The variables representing the times since the last relapse, disease duration until a DMT was administered, disease duration until a high-efficacy DMT was administered, and time since the last Fampridine administration were transformed according to an $1/(1 + t)$ scaling, with $t$ the actual time. If no time could be defined because, e.g., no DMT has ever been administered, the transformed variable was set to 0. If $t < 0$, which can happen because of erroneous dates in the dataset, the transformed variable was also set to 1.

The *longitudinal feature set* contains the dates and values for the following variables: all measured EDSS values and KFS scores, relapses occurrence (encoded as a binary variable set to 1 when a relapse occurs), relapse position (brainstem, pyramidal tract or other), cumulative relapse count, MS course, DMT administration (start and end dates), induction DMT administration (start date), and Fampridine administration. The timing of measurements was expected to be informative [17, 23].

## Models

The disability progression was framed as a classification problem. There exists a large literature on machine learning models for clinical applications [27–29]. The following models were used to predict disability progression: a multi-layer perceptron, a Bayesian neural network, and a temporal attention model with continuous temporal embeddings [30]. This work was supported by a large project (Flanders AI) and those models were selected as the best performing ones among a larger array of candidate models implemented by the different partners (see S1 Text for details). We followed the TRIPOD guidelines for reporting prognostic models [31]. The checklist can be found in Fig 8, at the end of this section.

The multi-layer perceptron model is a neural network architecture that takes as input the static and dynamic features set, represented as a fixed length vector. The model is composed of five hidden layers of dimension 128.

The Bayesian neural network has a similar architecture as the multi-layer perceptron, but provides uncertainty estimates on the weights of the last hidden layer by incorporating MCdropout [32]. This should confer better generalization capabilities as well as better calibration.

The temporal attention model relies on a transformer architecture [30]. In contrast to the previous models, this architecture is able to handle the longitudinal feature set, as it is able to process the whole clinical time series. Each visit is encoded as a fixed-length vector along with a mask for missing features and a continuous temporal embedding. This temporal embedding allows for arbitrary time differences between measurements, and is therefore especially suited for clinical time series where irregular sampling is most common. The static and the dynamic feature sets were included in the model as extra temporal features that are repeated over the patient history. Two temporal attention layers with dimension 128 were used. The code for training the models and the final models are publicly available and can be found at https://gitlab.com/edebrouwer/ms_benchmark.

## Evaluation

The dataset was split into 60% for training, 20% for validation and 20% for testing. The validation data was used to optimize the hyperparameters of the models. Post-hoc calibration methods (Platt scaling [33] and isotonic regression [34]) were used on the validation set and the performance evaluated on the test set.

TRIPOD Checklist: Prediction Model Development and Validation

| Section/Topic | Item | | Checklist Item | Page |
|---|---|---|---|---|
| **Title and abstract** | | | | |
| Title | 1 | D;V | Identify the study as developing and/or validating a multivariable prediction model, the target population, and the outcome to be predicted. | OK |
| Abstract | 2 | D;V | Provide a summary of objectives, study design, setting, participants, sample size, predictors, outcome, statistical analysis, results, and conclusions. | OK |
| **Introduction** | | | | |
| Background and objectives | 3a | D;V | Explain the medical context (including whether diagnostic or prognostic) and rationale for developing or validating the multivariable prediction model, including references to existing models. | OK |
| | 3b | D;V | Specify the objectives, including whether the study describes the development or validation of the model or both. | OK |
| **Methods** | | | | |
| Source of data | 4a | D;V | Describe the study design or source of data (e.g., randomized trial, cohort, or registry data), separately for the development and validation data sets, if applicable. | OK |
| | 4b | D;V | Specify the key study dates, including start of accrual; end of accrual; and, if applicable, end of follow-up. | OK |
| Participants | 5a | D;V | Specify key elements of the study setting (e.g., primary care, secondary care, general population) including number and location of centres. | OK |
| | 5b | D;V | Describe eligibility criteria for participants. | OK |
| | 5c | D;V | Give details of treatments received, if relevant. | OK |
| Outcome | 6a | D;V | Clearly define the outcome that is predicted by the prediction model, including how and when assessed. | OK |
| | 6b | D;V | Report any actions to blind assessment of the outcome to be predicted. | NA |
| Predictors | 7a | D;V | Clearly define all predictors used in developing or validating the multivariable prediction model, including how and when they were measured. | OK |
| | 7b | D;V | Report any actions to blind assessment of predictors for the outcome and other predictors. | NA |
| Sample size | 8 | D;V | Explain how the study size was arrived at. | OK |
| Missing data | 9 | D;V | Describe how missing data were handled (e.g., complete-case analysis, single imputation, multiple imputation) with details of any imputation method. | OK |
| Statistical analysis methods | 10a | D | Describe how predictors were handled in the analyses. | OK |
| | 10b | D | Specify type of model, all model-building procedures (including any predictor selection), and method for internal validation. | OK |
| | 10c | V | For validation, describe how the predictions were calculated. | OK |
| | 10d | D;V | Specify all measures used to assess model performance and, if relevant, to compare multiple models. | OK |
| | 10e | V | Describe any model updating (e.g., recalibration) arising from the validation, if done. | OK |
| Risk groups | 11 | D;V | Provide details on how risk groups were created, if done. | NA |
| Development vs. validation | 12 | V | For validation, identify any differences from the development data in setting, eligibility criteria, outcome, and predictors. | OK |
| **Results** | | | | |
| Participants | 13a | D;V | Describe the flow of participants through the study, including the number of participants with and without the outcome and, if applicable, a summary of the follow-up time. A diagram may be helpful. | OK |
| | 13b | D;V | Describe the characteristics of the participants (basic demographics, clinical features, available predictors), including the number of participants with missing data for predictors and outcome. | OK |
| | 13c | V | For validation, show a comparison with the development data of the distribution of important variables (demographics, predictors and outcome). | OK |
| Model development | 14a | D | Specify the number of participants and outcome events in each analysis. | OK |
| | 14b | D | If done, report the unadjusted association between each candidate predictor and outcome. | NA |
| Model specification | 15a | D | Present the full prediction model to allow predictions for individuals (i.e., all regression coefficients, and model intercept or baseline survival at a given time point). | OK |
| | 15b | D | Explain how to the use the prediction model. | OK |
| Model performance | 16 | D;V | Report performance measures (with CIs) for the prediction model. | OK |
| Model-updating | 17 | V | If done, report the results from any model updating (i.e., model specification, model performance). | NA |
| **Discussion** | | | | |
| Limitations | 18 | D;V | Discuss any limitations of the study (such as nonrepresentative sample, few events per predictor, missing data). | OK |
| Interpretation | 19a | V | For validation, discuss the results with reference to performance in the development data, and any other validation data. | OK |
| | 19b | D;V | Give an overall interpretation of the results, considering objectives, limitations, results from similar studies, and other relevant evidence. | OK |
| Implications | 20 | D;V | Discuss the potential clinical use of the model and implications for future research. | OK |
| **Other information** | | | | |
| Supplementary information | 21 | D;V | Provide information about the availability of supplementary resources, such as study protocol, Web calculator, and data sets. | OK |
| Funding | 22 | D;V | Give the source of funding and the role of the funders for the present study. | OK |

*Items relevant only to the development of a prediction model are denoted by D, items relating solely to a validation of a prediction model are denoted by V, and items relating to both are denoted D;V. We recommend using the TRIPOD Checklist in conjunction with the TRIPOD Explanation and Elaboration document.

**Fig 8. TRIPOD checklist.**

The test set was not seen during model training and hyperparameter optimization. To produce a measure of uncertainty of the performance of the models, the procedure of splitting the data and training the models was repeated five times, corresponding to five splits.

As the dataset consists of patients from different centers, we split the dataset such that the validation and test sets represent an external validation. Patients from the same centers were therefore assigned to the same set (training, validation or test).

Discrimination was evaluated using the area under the receiving operator characteristic (ROC-AUC) and the area under the precision recall curve (AUC-PR). Calibration was evaluated numerically using the Brier score and the expected calibration error (ECE) with 20 bins. Calibration was also evaluated visually using reliability diagrams.

The list of main hyperparameteres of each method, along with the values used for cross-validation are presented in S9, S10, S11, S12, S13 and S14 Tables.

## Tripod checklist

The design of the algorithms carefully followed the TRIPOD checklist as shown on Fig 8. All points were checked or were not applicable in our study. This consists of the following:

- 6b. Report any actions to blind assessment of the outcome to be predicted.

- 11. *Provide details on how risk groups were created, if done.* No risk groups were identified in this study.

- 14b. This can only be done for statistical models. However, we reported measures of variables importance.

- 17. *Model updating.* The models proposed here were not updates of previous iterations but rather their first development.

Note also that no sample size calculations were performed; the size of this retrospective dataset was fixed.

## Supporting information

**S1 Fig. ROC-AUC scores per MS center.** ROC-AUC of individual centers in the test set against the size of the center. As the size of the centers grows, the performance converges to the average ROC-AUC. As the size of centers shrinks, the variability in performance increases, which is statistically expected due to low sample size. Centers with no progression are not plotted (because ROC-AUC is not defined in this case).
(PDF)

**S2 Fig. Visualization of the different countries in the dataset.** Each country is represented as the set of vectors of static variables for each episode. A distance between countries was computed using earth mover distance. The 2D visualization was obtained by using multidimensional scaling (MDS).
(PNG)

**S3 Fig. Visualization of the different clinical centers in the dataset.** Each center is represented as the set of vectors of static variables for each episode. A distance between centers was computed using earth mover distance. The 2D visualization was obtained by using multidimensional scaling (MDS). We color each center by its country of origin.
(PNG)

**S4 Fig. Calibration diagram for all models.** Calibration curves of the different models on the test set (fold (*e.g.* train-test split) 0). Calibration was performed using Platt scaling [33]. A good calibration was observed for all models. The discrepancy with the ideal line (dotted) in the larger scores regime can be explained by the lowest number of data points in that region, leading to more variance.
(PDF)

**S5 Fig. Predicted percentage of worsening per subgroup.** Predicted percentage of worsening per subgroup, for both MS Courses and EDSS larger or smaller than 5.5. Green is the actual prevalence for the age groups on the x-axis, and red and purple are model predictions. This shows the calibration performance for different subgroups. An acceptable discrepancy is observed (of maximum 3 points), and a tendency of the models to underestimate the prevalence of disability progression.
(PDF)

**S1 Table. Summary statistics of the patients cohort.** Summary statistics of the cohort of interest after patient and sample selection. For all variables the value at the last recorded visit was used. KFS stands for Kurtzke Functional Systems Score, DMT for Disease Modifying Therapy, CIS for Clinically Isolated Syndrome.
(PDF)

**S2 Table. Summary statistics of the performance measures (Cohort with minimum 3 visits).** ROC-AUC, AUC-PR, Brier Score and ECE of all models (averages ± standard deviations). Cohort of patients with a least 3 visits with EDSS in the last 3.25 years.
(PDF)

**S3 Table. Summary statistics of the performance measures (Cohort with minimum 6 visits).** ROC-AUC, AUC-PR, Brier Score and ECE of all models (averages ± standard deviations). Cohort of patients with a least 6 visits with EDSS in the last 3.25 years.
(PDF)

**S4 Table. Summary statistics of the performance measures on different MS subgroups (Cohort with minimum 3 visits).** ROC-AUC, AUC-PR, Brier Score and ECE of all models on the different MS course subgroups (averages ± standard deviations). Primary Progressive (PP), Relapsing Remitting (RR) and Secondary Progressive are considered (SP). Cohort of patients with a least 3 visits with EDSS in the last 3.25 years.
(PDF)

**S5 Table. Summary statistics of the performance measures on different MS subgroups (Cohort with minimum 6 visits).** ROC-AUC, AUC-PR, Brier Score and ECE of all models on the different MS course subgroups (averages ± standard deviations). Primary Progressive (PP), Relapsing Remitting (RR) and Secondary Progressive are considered (SP). Cohort of patients with a least 6 visits with EDSS in the last 3.25 years.
(PDF)

**S6 Table. Summary statistics of the performance measures on different severity subgroups (Cohort with minimum 3 visits).** ROC-AUC, AUC-PR, Brier Score and ECE by severity subgroup (averages ± standard deviations). Low severity patients are defined as the ones with EDSS ≤ 5.5 at baseline, while high severity patients are defined as having EDSS > 5.5 at baseline. Cohort of patients with a least 3 visits with EDSS in the last 3.25 years.
(PDF)

**S7 Table. Summary statistics of the performance measures on different severity subgroups (Cohort with minimum 6 visits).** ROC-AUC, AUC-PR, Brier Score and ECE by severity subgroup (averages ± standard deviations). Low severity patients are defined as the ones with EDSS ≤ 5.5 at baseline, while high severity patients are defined as having EDSS > 5.5 at baseline. Cohort of patients with a least 6 visits with EDSS in the last 3.25 years.
(PDF)

**S8 Table. Features importance for different performance metrics.** Features are ranked by order of importance for the Dynamic Model. Feature importance is assessed by the average difference in performance when the specific feature is shuffled. Averages ± standard deviations are reported.
(PDF)

**S9 Table. Hyperparameters table for the temporal attention model.** List of hyperparameters used for training the models.
(PDF)

**S10 Table. Hyperparameters table for the multi-layer perceptron model.** List of hyperparameters used for training the models.
(PDF)

**S11 Table. Hyperparameters table for the recurrent neural network model.** List of hyperparameters used for training the models.
(PDF)

**S12 Table. Hyperparameters table for the dynamic MTP model.** List of hyperparameters used for training the models.
(PDF)

**S13 Table. Hyperparameters table for the factorization machines model.** List of hyperparameters used for training the models.
(PDF)

**S14 Table. Hyperparameters table for the logistic regression model.** List of hyperparameters used for training the models.
(PDF)

**S1 Text. Models description.** Description of the Bayesian neural networks, DeepMTP, and Factorization Machines models.
(PDF)

## Acknowledgments

The authors also wish to acknowledge the MSBase contributors for sharing the clinical data:

- Eva Kubala Havrdova, Charles University in Prague and General University Hospital, Prague, Czech Republic
- Serkan Ozakbas, Dokuz Eylul University, Konak/Izmir, Turkey
- Marco Onofrj, University G. d'Annunzio, Chieti, Italy
- Raed Alroughani, Amiri Hospital, Sharq, Kuwait
- Maria Pia Amato, University of Florence, Florence, Italy

- Katherine Buzzard, Box Hill Hospital, Melbourne, Australia

- Cavit Boz, KTU Medical Faculty Farabi Hospital, Trabzon, Turkey

- Vahid Shaygannejad, Isfahan University of Medical Sciences, Isfahan, Iran

- Jens Kuhle, Universitatsspital Basel, Basel, Switzerland

- Bassem Yamout, American University of Beirut Medical Center, Beirut, Lebanon

- Recai Turkoglu, Haydarpasa Numune Training and Research Hospital, Istanbul, Turkey

- Julie Prevost, CSSS Saint-Jérôme, Saint-Jerome, Canada

- Ernest Butler, Monash Medical Centre, Melbourne, Australia

- Celia Oreja-Guevara, Hospital Clinico San Carlos, Madrid, Spain

- Richard Macdonell, Austin Health, Melbourne, Australia

- Ricardo Fernandez Bolaños, Hospital Universitario Virgen de Valme, Seville, Spain

- Marie D'hooghe, Nationaal MS Centrum, Melsbroek, Belgium

- Liesbeth Van Hijfte, Universitary Hospital Ghent, Ghent, Belgium

- Helmut Butzkueven, The Alfred Hospital, Melbourne, Australia

- Michael Barnett, Brain and Mind Centre, Sydney, Australia

- Justin Garber, Westmead Hospital, Sydney, Australia

- Sarah Besora, Hospital Universitari MútuaTerrassa, Barcelona, Spain

- Edgardo Cristiano, Centro de Esclerosis Múltiple de Buenos Aires (CEMBA), Buenos Aires, Argentina

- Magd Zakaria, Ain Shams University

- Maria Laura Saladino, INEBA—Institute of Neuroscience Buenos Aires, Buenos Aires, Argentina

- Shlomo Flechter, Assaf Harofeh Medical Center, Beer-Yaakov, Israel

- Leontien Den braber-Moerland, Francicus Ziekenhuis, Roosendaal, Netherlands

- Fraser Moore, Jewish General Hospital, Montreal, Canada

- Rana Karabudak, Hacettepe University, Ankara, Turkey

- Claudio Gobbi, Ospedale Civico Lugano, Lugano, Switzerland

- Jennifer Massey, St Vincent's Hospital, Sydney, Australia

- Nevin Shalaby, Kasr Al Ainy MS research Unit (KAMSU), Cairo, Egypt

- Jabir Alkhaboori, Royal Hospital, Muscat, Oman

- Cameron Shaw, Geelong Hospital, Geelong, Australia

- Jose Andres Dominguez, Hospital Universitario de la Ribera, Alzira, Spain

- Jan Schepel, Waikato Hospital, Hamilton, New Zealand

- Krisztina Kovacs, Péterfy Sandor Hospital, Budapest, Hungary

- Pamela McCombe, Royal Brisbane and Women's Hospital, Brisbane, Australia

- Bhim Singhal, Bombay Hospital Institute of Medical Sciences, Mumbai, India

- Mike Boggild, Townsville Hospital, Townsville, Australia

- Imre Piroska, Veszprém Megyei Csolnoky Ferenc Kórház zrt., Veszprem, Hungary

- Neil Shuey, St Vincents Hospital, Fitzroy, Melbourne, Australia

- Carlos Vrech, Sanatorio Allende, Cordoba, Argentina

- Tatjana Petkovska-Boskova, Clinic of Neurology Clinical Center, Skopje, Macedonia

- Ilya Kister, New York University Langone Medical Center, New York, United States

- Cees Zwanikken, University Hospital Nijmegen, Nijmegen, Netherlands

- Jamie Campbell, Craigavon Area Hospital, Craigavon, United Kingdom

- Etienne Roullet, MS Clinic, Hopital Tenon, Paris, France

- Cristina Ramo-Tello, Hospital Germans Trias i Pujol, Badalona, Spain

- Jose Antonio Cabrera-Gomez, Centro Internacional de Restauracion Neurologica, Havana, Cuba

- Maria Edite Rio, Centro Hospitalar Universitario de Sao Joao, Porto, Portugal

- Pamela McCombe, University of Queensland, Brisbane, Australia

- Mark Slee, Flinders University, Adelaide, Australia

- Saloua Mrabet, Razi Hospital, Manouba, Tunisia

## Author Contributions

**Conceptualization:** Edward De Brouwer, Thijs Becker, Lorin Werthen-Brabants, Guy Laureys, Bart Van Wijmeersch, Veronica Popescu, Guy Laureys, Yves Moreau, Liesbet Peeters.

**Data curation:** Edward De Brouwer, Thijs Becker, Dana Horakova, Francesco Patti, Guillermo Izquierdo, Sara Eichau, Marc Girard, Alexandre Prat, Alessandra Lugaresi, Pierre Grammond, Tomas Kalincik, Raed Alroughani, Francois Grand'Maison, Olga Skibina, Murat Terzi, Jeannette Lechner-Scott, Oliver Gerlach, Samia J. Khoury, Elisabetta Cartechini, Vincent Van Pesch, Maria José Sà, Bianca Weinstock-Guttman, Yolanda Blanco, Radek Ampapa, Daniele Spitaleri, Claudio Solaro, Davide Maimone, Aysun Soysal, Gerardo Iuliano, Riadh Gouider, Tamara Castillo-Triviño, José Luis Sánchez-Menoyo, Anneke van der Walt, Jiwon Oh, Eduardo Aguera-Morales, Ayse Altintas, Abdullah Al-Asmi, Koen de Gans, Yara Fragoso, Tunde Csepany, Suzanne Hodgkinson, Norma Deri, Talal Al-Harbi, Bruce Taylor, Orla Gray, Patrice Lalive, Csilla Rozsa, Chris McGuigan, Allan Kermode, Angel Pérez Sempere, Simu Mihaela, Magdolna Simo, Todd Hardy, Danny Decoo, Stella Hughes, Nikolaos Grigoriadis, Attila Sas, Norbert Vella.

**Formal analysis:** Edward De Brouwer, Thijs Becker, Lorin Werthen-Brabants, Pieter Dewulf, Dimitrios Iliadis.

**Funding acquisition:** Liesbet Peeters.

**Investigation:** Edward De Brouwer, Thijs Becker.

**Methodology:** Edward De Brouwer, Thijs Becker, Yves Moreau.

**Project administration:** Liesbet Peeters.

**Software:** Edward De Brouwer, Thijs Becker, Lorin Werthen-Brabants, Pieter Dewulf, Dimitrios Iliadis.

**Supervision:** Yves Moreau, Liesbet Peeters.

**Validation:** Thijs Becker, Guy Laureys.

**Writing – original draft:** Edward De Brouwer, Thijs Becker, Lorin Werthen-Brabants, Pieter Dewulf, Dimitrios Iliadis, Yves Moreau, Liesbet Peeters.

**Writing – review & editing:** Edward De Brouwer, Thijs Becker, Lorin Werthen-Brabants, Pieter Dewulf, Dimitrios Iliadis, Cathérine Dekeyser, Guy Laureys, Bart Van Wijmeersch, Veronica Popescu, Tom Dhaene, Dirk Deschrijver, Willem Waegeman, Bernard De Baets, Michiel Stock, Yolanda Blanco, Guy Laureys, Yves Moreau, Liesbet Peeters.

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
