## [Decision Letter · Decision Letter 0]

21 Aug 2023

PDIG-D-23-00247

Machine-learning-based prediction of disability progression in multiple sclerosis: an observational, international, multi-center study

PLOS Digital Health

Dear Dr. De Brouwer,

Thank you for submitting your manuscript to PLOS Digital Health. After careful consideration, we feel that it has merit but does not fully meet PLOS Digital Health's publication criteria as it currently stands. Therefore, we invite you to submit a revised version of the manuscript that addresses the points raised during the review process.

Please submit your revised manuscript within 60 days Oct 20 2023 11:59PM. If you will need more time than this to complete your revisions, please reply to this message or contact the journal office at digitalhealth@plos.org. Please include the following items when submitting your revised manuscript:

We look forward to receiving your revised manuscript.

Kind regards,

Ryan S McGinnis

Academic Editor

PLOS Digital Health

Journal Requirements:

1. We ask that a manuscript source file is provided at Revision. Please upload your manuscript file as a .doc, .docx, .rtf or .tex.

2. Please provide separate figure files in .tif or .eps format only and remove any figures embedded in your manuscript file. Please also ensure that all files are under our size limit of 10MB.

Additional Editor Comments (if provided):

Reviewers' comments:

Reviewer's Responses to Questions

**Comments to the Author**

1. Does this manuscript meet PLOS Digital Health’s publication criteria? Is the manuscript technically sound, and do the data support the conclusions? The manuscript must describe methodologically and ethically rigorous research with conclusions that are appropriately drawn based on the data presented.

Reviewer #1: Partly

Reviewer #2: Partly

Reviewer #3: Yes

2. Has the statistical analysis been performed appropriately and rigorously?

Reviewer #1: I don't know

Reviewer #2: Yes

Reviewer #3: Yes

3. Have the authors made all data underlying the findings in their manuscript fully available (please refer to the Data Availability Statement at the start of the manuscript PDF file)?

Reviewer #1: Yes

Reviewer #2: Yes

Reviewer #3: Yes

4. Is the manuscript presented in an intelligible fashion and written in standard English?

Reviewer #1: Yes

Reviewer #2: Yes

Reviewer #3: Yes

5. Review Comments to the Author

Reviewer #1: The authors present a machine learning method to predict whether a disease progression will occur in persons with MS. They present numerous algorithms and train them on commonly available features. Overall the paper presents meaningful findings, however, the framing of the paper and claims are not supported by the results.

Major Concerns:

- The authors claim to train models 'for disability progression prediction in MS.' While this may be the end goal of the work, the work presented in this manuscript is a classification of whether a change will occur or not, not a prediction of the actual disease progression. The presented work is an important stepping stone, however, it cannot address the motivation suggests for uses of patient planning and care. Please reframe the paper in a more appropriate manner to reflect the results presented.

- The exclusion criteria limit those who are newly diagnosed and/or currently have a steady disease state. This is partially addressed as a limitation in the discussion but this needs to be expanded. These exclusions limit the model to predict whether a change will occur in those who already present changes instead of PwMS overall. 

- P.18 is this definition of disease progression a clinical standard or is it introduced for this work? Need to define and motivate or cite. 

- Please provide more details on model training and evaluation (eg. training epochs)

- Several models are presented with very similar results and claims are made about which is best. Please add significance tests to determine if these differences are significant. 

Minor Concerns:

- Throughout the manuscript there is a mix of present and past tense

- Define all acronyms before they are used in text

- Need to include definitions of acronyms in table captions

- Define all elements of equations clearly (e.g. page 26; Parameter p seems to be defined in figured but not text)

- Some results, such as table 5, may be easier to interpret as a figure/bar chart

- Please fix reference to supplementary materials on page 24 line 329 ?? and corresponding table in supplementary materials

Reviewer #2: General:

This manuscript shows the ability ofpredicting MS progression based on a large data set. The study design is superior in its choice and rational for the chosen input variables which all target the clinical feasability. The manuscript comes with some flaws in writing; the methods section is narrative in major parts, the structure is confusing and the different names for the used models lack explanation. This stands in contrast to being applicable in clinical practise. Any clinician that will read the paper, will stop reading the methods by being confused, which would be a pity. 

The discussion is missing as there is not a single reference to other pulished models. In its current status, the paper needs to be rewritten in large parts to be comprehensive.

Introduction:

The introduction stays vague in developing the rationale for the parameters of interest. Also, no hypotheses are formulated. Instead, the results are already summarized which feels like a repetition of the abstract.

Results:

The Cohort section of the results is mainly methods, I would therefore suggest removing this information.

Tables: please explain all the abbreviations. Also the caption for the tables are ususaly above the table, of not requested differently.

Discussion:

Was the model also tested on less input span? Would be interesting what the minimal number of required monitoring points would be.

The discussion has not a single reference to other published models (that were named in the introduction). Please report and compare the results to existing prediction models to give the reader a fair chance to evaluate the reported outcome. 

Methods:

General:

Although I appreciate a detailed description of the methods, the amount of information is just overwhelming. Describing why a parameter is abbreviated is too much (e.g. why w is choosen to abbreviate worsening). The methods are too narrative and wouldbe shorter with lessstory telling. Also, the methods are quite chaotic. 

If the section about valid/non-valid samples would be clearer, Figure 5 would not be needed. 

Line 240 ff: why were the two cohorts defined? There is so much description of things that are not important, but then this is not clear.

What was the rational behind training the model with these long time spans? For an efficient patient management, much shorter tie frames are required. Also, which real predictive value has the model, if so much input is needed?

Line 291: 'If data can be missing...' probaby better to use 'If data was missing...'

Line 330: supplementary material number is not valid.

It is completely unclear to me why first 3 different models are named and described briefly, followed by a section where they are again explained in great detail but differently named and in a different order then the summary description. any reader that is not deeply informed about machine learning will be lost. I encourage the authors to be concise and precise.

Reviewer #3: While the authors present a useful application of standard machine learning methods and architectures in the clinical setting of MS, the novelty of the approaches used by the authors is somewhat limited. I would recommend the authors consider the following:

1. The authors claim to evaluate the performance of their model on an external dataset (where the "externalization" refers to splitting the data by patient center). Have the authors conducted any data-distribution level visualizations/statistics to verify whether the data occupy different distributions? Specifically, have the authors considered other axes of data heterogeneity in addition to patient population?

2. It is great that the authors assessed the degree of model calibration, together with the Brier score, in their work. Given the size of the dataset, have the authors considered the degree to which model predictions are repeatable across patients and/or explainable?

6. PLOS authors have the option to publish the peer review history of their article (what does this mean?). If published, this will include your full peer review and any attached files.

**Do you want your identity to be public for this peer review?** For information about this choice, including consent withdrawal, please see our Privacy Policy.

Reviewer #1: No

Reviewer #2: No

Reviewer #3: No

---

## [Decision Letter · Decision Letter 1]

27 Feb 2024

PDIG-D-23-00247R1

Machine-learning-based prediction of disability progression in multiple sclerosis: an observational, international, multi-center study

PLOS Digital Health

Dear Dr. De Brouwer,

Thank you for submitting your manuscript to PLOS Digital Health. After careful consideration, we feel that it has merit but does not fully meet PLOS Digital Health's publication criteria as it currently stands. Therefore, we invite you to submit a revised version of the manuscript that addresses the points raised during the review process.

Please submit your revised manuscript within 60 days Apr 27 2024 11:59PM. If you will need more time than this to complete your revisions, please reply to this message or contact the journal office at digitalhealth@plos.org. Please include the following items when submitting your revised manuscript:

We look forward to receiving your revised manuscript.

Kind regards,

Ryan S McGinnis

Academic Editor

PLOS Digital Health

Journal Requirements:

Additional Editor Comments (if provided):

Thank you for your revised submission. We have arranged review of this revised manuscript. Please address the remaining comments in your revision.

Reviewers' comments:

Reviewer's Responses to Questions

**Comments to the Author**

1. If the authors have adequately addressed your comments raised in a previous round of review and you feel that this manuscript is now acceptable for publication, you may indicate that here to bypass the “Comments to the Author” section, enter your conflict of interest statement in the “Confidential to Editor” section, and submit your "Accept" recommendation.

Reviewer #1: All comments have been addressed

Reviewer #4: (No Response)

2. Does this manuscript meet PLOS Digital Health’s publication criteria? Is the manuscript technically sound, and do the data support the conclusions? The manuscript must describe methodologically and ethically rigorous research with conclusions that are appropriately drawn based on the data presented.

Reviewer #1: Partly

Reviewer #4: Partly

3. Has the statistical analysis been performed appropriately and rigorously?

Reviewer #1: Yes

Reviewer #4: Yes

4. Have the authors made all data underlying the findings in their manuscript fully available (please refer to the Data Availability Statement at the start of the manuscript PDF file)?

Reviewer #1: Yes

Reviewer #4: Yes

5. Is the manuscript presented in an intelligible fashion and written in standard English?

Reviewer #1: Yes

Reviewer #4: Yes

6. Review Comments to the Author

Reviewer #1: Thank you for addressing the comments. The additional details and clarifications are greatly appreciated. Based on the pvalues added in table 2, there is not sufficient evidence that the attention model performs best. Please adjust the wording throughout (including introduction) to address this. Additionally, there are a few formatting (eg. p7 line ~52 line break) and wording errors that could use one more readthrough.

Reviewer #4: This manuscript presents a machine learning approach for predicting if a person with multiple sclerosis will experience disability progression in the next two years based on data commonly collected during routine clinical care. Models were trained on a very large, multi-national dataset (MSBase). I was not a reviewer on the prior version of the manuscript, but I have reviewed the comments from prior reviewers and believe that the authors have adequately addressed many of their concerns. In my review, I have identified several areas (below) that would benefit from revision to strengthen this manuscript and further highlight this important work. 

-Abstract, Methods: typo – expended -> expanded

-line64: What is the scientific or clinical rationale for predicting progression in the next two years? Why not 1 year? Or 6 months? This seems to have also been a question from the prior reviewers that has not been addressed and would be helpful for readers. 

-lines 114-115: it is noted that the lower performance observed in in the primary progressive and secondary progressive subgroups was due to small sample size, but how is that known? Was this tested in some way? 

-tables 2-4: it would be helpful to note, in the caption, what is being reported after the +/- in these tables. 

-tables 3-4: an indication of the training set size for each model being compared would be helpful as it is not exactly clear with all of the subgroups.

-lines 153-154: While it is argued that the presented models are a ‘significant advance towards deploying AI in clinical practice in MS’ - how do we know what good enough performance is in this context? Is a ROC AUC of 0.72 and PR-AUC of 0.26 sufficient for predicting MS progression? What are the negative byproducts of an incorrect prediction (both false positive and negative)? It would be helpful for readers if these results could be further evaluated and placed in their intended context of use in the discussion section. 

-Discussion: this section would be strengthened with a more detailed discussion of the reported results. For example, the results of Tables 3 and 4 are clearly important, but are not discussed. It would be helpful for the authors to further elucidate what may be causing the observed differences in model performance and to highlight what those performance differences may imply for the translation of this approach into the clinical environment. 

-there seems to be an extra (2) between lines 271 and 272

-line323: 11.64% progression events suggests a pretty significant imbalance in the data, but it is not clear how this was dealt with in training and evaluating the models. This should be described more clearly.

7. PLOS authors have the option to publish the peer review history of their article (what does this mean?). If published, this will include your full peer review and any attached files.

**Do you want your identity to be public for this peer review?** For information about this choice, including consent withdrawal, please see our Privacy Policy. 

Reviewer #1: No

Reviewer #4: No

---

## [Decision Letter · Decision Letter 2]

14 May 2024

Machine-learning-based prediction of disability progression in multiple sclerosis: an observational, international, multi-center study

PDIG-D-23-00247R2

Dear Dr De Brouwer,

We are pleased to inform you that your manuscript 'Machine-learning-based prediction of disability progression in multiple sclerosis: an observational, international, multi-center study' has been provisionally accepted for publication in PLOS Digital Health.

Best regards,

Ryan S McGinnis

Academic Editor

PLOS Digital Health

Many thanks to the authors for addressing the remaining reviewer comments.

Reviewer Comments (if any, and for reference):

Reviewer's Responses to Questions

**Comments to the Author**

1. If the authors have adequately addressed your comments raised in a previous round of review and you feel that this manuscript is now acceptable for publication, you may indicate that here to bypass the “Comments to the Author” section, enter your conflict of interest statement in the “Confidential to Editor” section, and submit your "Accept" recommendation.

Reviewer #1: All comments have been addressed

2. Does this manuscript meet PLOS Digital Health’s publication criteria? Is the manuscript technically sound, and do the data support the conclusions? The manuscript must describe methodologically and ethically rigorous research with conclusions that are appropriately drawn based on the data presented.

Reviewer #1: Yes

3. Has the statistical analysis been performed appropriately and rigorously?

Reviewer #1: Yes

4. Have the authors made all data underlying the findings in their manuscript fully available (please refer to the Data Availability Statement at the start of the manuscript PDF file)?

Reviewer #1: Yes

5. Is the manuscript presented in an intelligible fashion and written in standard English?

Reviewer #1: Yes

6. Review Comments to the Author

Reviewer #1: Comments addressed. Thank you.

7. PLOS authors have the option to publish the peer review history of their article (what does this mean?). If published, this will include your full peer review and any attached files.

**Do you want your identity to be public for this peer review?** For information about this choice, including consent withdrawal, please see our Privacy Policy.

Reviewer #1: No
